# Cell-type-specific genomics reveals histone modification dynamics in mammalian meiosis

Kwan-Wood Gabriel Lam [iD] [1], Kevin Brick [iD] [1], Gang Cheng[1], Florencia Pratto[1] & R. Daniel Camerini-Otero[1]

Meiosis is the specialized cell division during which parental genomes recombine to create genotypically unique gametes. Despite its importance, mammalian meiosis cannot be studied in vitro, greatly limiting mechanistic studies. In vivo, meiocytes progress asynchronously through meiosis and therefore the study of specific stages of meiosis is a challenge. Here, we describe a method for isolating pure sub-populations of nuclei that allows for detailed study of meiotic substages. Interrogating the H3K4me3 landscape revealed dynamic chromatin transitions between substages of meiotic prophase I, both at sites of genetic recombination and at gene promoters. We also leveraged this method to perform the first comprehensive, genome-wide survey of histone marks in meiotic prophase, revealing a heretofore unappreciated complexity of the epigenetic landscape at meiotic recombination hotspots. Ultimately, this study presents a straightforward, scalable framework for interrogating the complexities of mammalian meiosis.

[1] Genetics and Biochemistry Branch, National Institute of Diabetes, Digestive and Kidney Diseases, National Institutes of Health, Bethesda, MD 20892, USA. Correspondence and requests for materials should be addressed to R.D.C-O. (email: rdcamerini@mail.nih.gov)

Meiosis is the specialized cell division required to form gametes. In meiosis the DNA of a parent cell is replicated, then haploid gametes are produced through two successive cell divisions. Distinct mechanisms are required to assure that chromosomes segregate accurately in each division and these events take place through a staged cascade of nuclear events. Unique to meiosis is the formation of programmed DNA double-strand breaks (DSBs) that result in recombination between parental haplotypes and whose repair tethers homologous chromosomes at the first meiotic division. Meiotic DSB formation, repair and recombination occur in a continuum of substages (leptonema, zygonema, pachynema, and diplonema) known collectively as meiotic prophase I (MPI), and each stage is characterized by specific nuclear events[1]. Despite its importance, technical challenges preclude the isolation and study of specific meiotic substages and many of the molecular mechanisms of meiosis remain poorly studied in higher eukaryotes. This is largely because mammalian meiosis cannot be adequately recapitulated in vitro[2]. In vivo, meiosis either occurs in a difficult-to-isolate niche, such as the fetal ovary, or in an asynchronously dividing tissue such as in adult testis.

Extant methods to study substages of meiotic prophase have been dominated by the use of crude enrichment strategies. One common approach has been to study spermatocytes in juvenile mice, where the first wave of meiosis progresses relatively synchronously[3,4]. Nonetheless, most cells in juvenile testis are not at the stage of interest because this synchrony is not absolute, and there are many non-meiotic cells. Furthermore, differences between the first and the subsequent rounds of spermatogenesis[5–7] may preclude generalization to adults. Chemically induced synchronization of meiosis has recently been developed[8]; however, the application is technically cumbersome and the consequences of chemical treatment have not been explored. An alternative to enrichment is the isolation of meiotic substages. Fluorescence-activated cell sorting (FACS) or sedimentation-based methods have been used to isolate particular substages[9–11]. However, since these methods rely on imprecise metrics such as cell size and chromatin content, the isolation and distinction between many substages is challenging. Thus, isolation-based approaches have not been broadly adapted to the study of the dynamic molecular events during meiosis.

We have devised a straightforward, yet highly specific strategy to isolate pure subpopulations of meiotic nuclei. This approach uses a variant of FACS[12] in which antibodies to intra-nuclear markers are used to select nuclei of interest. The proteins present in different meiotic substages are well studied[13–16] and we leverage this knowledge to isolate nuclei in which specific combinations of proteins are expressed. This allowed for the isolation of meiotic subpopulations ranging from 74 to 96% purity. Nuclei are fixed before sorting, which preserves in vivo interactions and prevents degradation during sample processing. Importantly, this fixation also means our method is the first to allow for downstream interrogation of genome-wide protein−DNA interactions in meiotic substages.

Although comprehensively studied in somatic cells, little is known about histone modification dynamics in meiotic cells. A case in point are the histone modifications that precede meiotic DSB formation, an early event in meiotic recombination. Programmed DSBs are targeted to sites defined by DNA binding and subsequent H3K4/K36-trimethylation by the PRDM9 protein[17,18]. PRDM9 gene expression[19], nuclear protein levels of PRDM9[20] and microscopy-based estimates of DSB formation[21,22] all imply that PRDM9 marks sites for DSB formation at the onset of MPI.

Using chromatin immunoprecipitation followed by high-throughput sequencing (ChIP-Seq) on sorted populations of nuclei, we determined that Histone H3 Lysine 4 trimethylation (H3K4me3) varies from stage-to-stage at meiotic DSB hotspots, gene promoters and a vast number of heretofore unannotated sites. These are the first genome-wide analyses of the dynamic chromatin landscape in meiosis. Also, we found that the histone code at DSB hotspots is more complex than has been previously appreciated[17,18], with multiple histone tail modifications exhibiting enrichment. This includes histone acetylation marks unlikely to have been deposited by the DSB-defining histone methyltransferase PRDM9.

Together, these data demonstrate the simplicity, utility and feasibility of our approach. This method opens the door to a far better understanding of the molecular mechanisms underlying meiosis and other complex developmental processes.

## Results

**Isolation of stage-specific spermatocyte nuclei**. We first devised a discriminative panel of intra-nuclear proteins to distinguish between five, classically defined MPI substages: leptonema, zygonema, early pachynema, late pachynema and diplonema (Fig. 1). Nuclei were extracted from formaldehyde-fixed testes from adult male mice, then incubated with antibodies against the desired proteins (see Methods). Nuclei were also stained with DAPI to allow measurement of DNA content. The combinatorial signals from these fluorescent markers were subsequently used to isolate specific subpopulations of spermatocyte nuclei using FACS (Fig. 2a, b; see Methods).

We used the meiosis-specific protein SCP3 to distinguish post-replicative meiotic prophase nuclei (4C) from fully replicated mitotic nuclei (also 4C). We then combined SCP3 with other markers for FACS. Our combination of antibodies required two separate sorts; one to isolate leptotene nuclei and a second to isolate the other four substages (Fig. 2a, b). In male mice, STRA8 regulates the entry to meiotic S phase, and is expressed from primitive spermatogonia to leptotene spermatocytes[16]. We therefore used STRA8 to isolate leptotene nuclei from later stages. To isolate the four other meiotic stages, we used the presence or absence of H1t and quantitative differences in SCP1 intensity (Figs. 1, 2a). H1t (testis-specific histone H1) is present from mid pachynema through MPI[23], whereas SCP1 is a marker of progressive chromosome synapsis from zygotene to pachytene stage and the SCP1 signal is diminished in diplonema when chromosomes desynapse[14] (Fig. 1). Thus, we collected seven populations, assessed the purity of each population and determined the gating thresholds for the four meiotic substages (Fig. 2b, c).

The purity of each sorted population was assessed using immunofluorescence microscopy to stage the nuclei according to the known localization of markers (Figs. 1, 2a, c, Supplementary Fig. 1, see Methods)[13–16]. The purity of the sorted leptotene population was 94.6–97.9% ($n = 2$), a >50-fold increase in purity compared to the percentage of leptotene cells in whole-testis (1.6–1.8% ($n = 2$), Fig. 2c). The purity of the other four populations ranged from 73.9 to 90.4%, each more than 20-fold enriched relative to the starting population from whole-testis (Fig. 2c). The least pure sample was that of zygotene (71.3–78.7% ($n = 3$)), likely because we rely entirely on quantitative differences in SCP1 to distinguish between adjacent stages (Figs. 1, 2b, c). Narrowing of the gate can increase purity but at the cost of yield (data not shown). Quantitative thresholding issues also explain the 9.0–16.9% ($n = 3$) of zygotene nuclei found in the sorted early pachytene population (Figs. 1, 2b, c).

From a single adult mouse, we obtained between 115,000 and 400,000 nuclei for each meiotic substage (Fig. 2d). This range broadly reflects differences in the proportion of each substage in the adult testis, where leptotene and zygotene nuclei are less

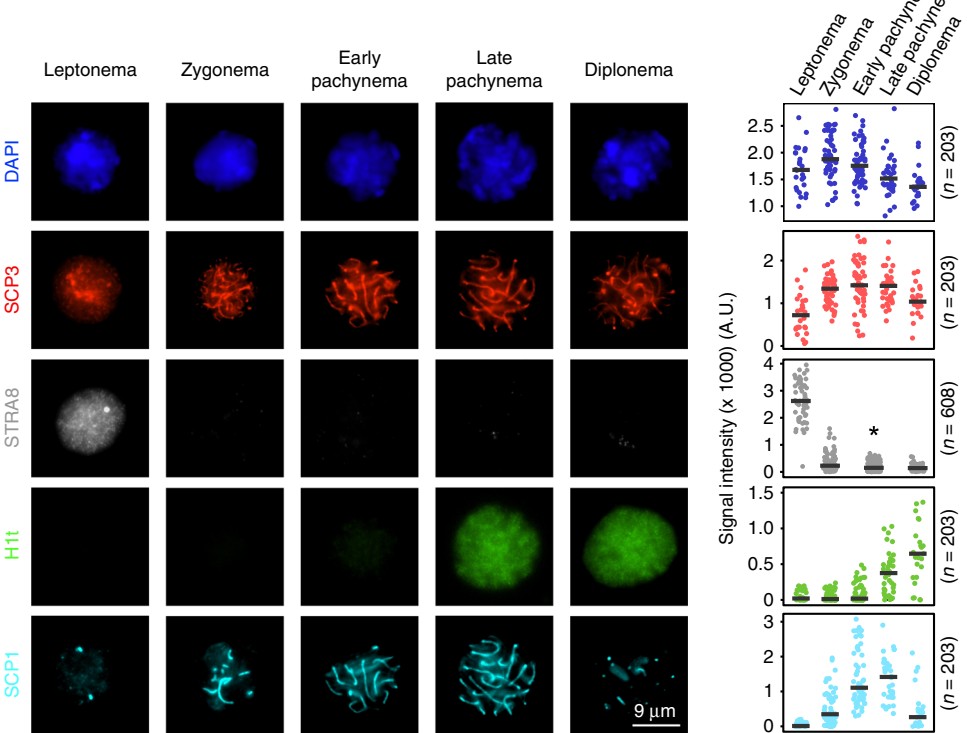

**Fig. 1** Immunofluorescence staining of spermatocyte nuclei. Immunofluorescence images and signal quantification of stage-specific spermatocyte nuclei through meiosis prophase I. Details for signal quantification are described in Methods. Microscopic images are selected from two independent experiments in which two different combinations of primary antibodies are used; one using SCP3, H1t and SCP1, another one using SCP3 and STRA8. *Early and late pachytene nuclei cannot be unambiguously differentiated in the absence of H1t staining, and are therefore merged for counting and signal quantification. Source data are provided as a Source Data file

abundant. Thus, the different substages are isolated in sufficient numbers for experiments using standard genomics protocols[24].

**Stage-specific ChIP-Seq reveals histone modification dynamics.** We next used ChIP-Seq to examine the dynamics of H3K4me3 in MPI. H3K4me3 is of key biological interest in meiosis because it regulates multiple independent dynamic processes during MPI; H3K4me3 is found at gene promoters and its presence correlates with active transcription[25,26], whereas PRDM9-mediated H3K4me3 marks the future sites of meiotic DSBs genome-wide[27,28]. We therefore performed H3K4me3 ChIP-Seq in the five aforementioned sorted populations to assess the temporal dynamics of this histone mark in MPI (Table 1).

We performed ChIP-Seq for H3K4me3 using a highly specific antibody (EpiCypher, 13-0028) (Supplementary Fig. 2). To allow for quantitative cross-comparison of H3K4me3 signals from samples of differing quality, we used a spike-in control containing H3K4me3-modified nucleosomes[29] (see Methods). We observed a highly dynamic H3K4me3 signal at hotspots in MPI; the H3K4me3 signal first appears at leptonema, and is maximal at zygonema, and gone by early pachynema (Fig. 3a, b). The maximal signal at zygonema was somewhat surprising because DSBs are formed and nuclear PRDM9 protein levels are maximal at leptonema[20]. Intriguingly, at strong hotspots, H3K4me3 is relatively weak at zygonema compared to leptonema (Supplementary Fig. 3). This implies that DSB formation or repair removes H3K4me3-modified nucleosomes, similar to findings in *Saccharomyces cerevisae*[30].

Most H3K4me3 ChIP-Seq studies, including all studies of H3K4me3 in meiosis have used antibodies that extensively cross-react with H3K4me2[31] (Supplementary Fig. 2). To allow cross-

comparison with published data we performed ChIP-Seq with the most commonly used antibody (Abcam ab8580[31]) and observed a broadly similar dynamic of histone modification at hotspots as compared to the specific antibody (Supplementary Fig. 4); the hotspot signal is still maximal at zygotene, but the nonspecific antibody gives a higher leptotene signal and a residual signal in early pachynema. We hypothesize that these differences reflect dynamic changes in H3K4 dimethylation at hotspots.

The H3K4me3 level at gene promoters is correlated with gene expression in mitotic cells[26]. We therefore investigated if the temporal dynamics of H3K4me3 at transcription start sites (TSSs) is predictive of meiotic gene expression patterns. Stage-specific gene expression in MPI has been deduced from classical cell sorting of meiotic cells followed by RNA-Seq[11,32]. We found that the H3K4me3 profiles at TSSs were positively correlated with gene expression through MPI (Spearman's $R = 0.27$; see Methods). This correlation was highly significant, far higher than the maximum correlation observed in randomized comparisons (empirical $P < 0.0001$; $N = 10,000$ bootstraps, $R_{max} = 0.03$, see Methods). To explore the relationship between H3K4me3 and gene expression in more detail, we clustered transcripts based on the temporal H3K4me3 pattern (Fig. 3c; optimal clusters = 5; $k$-means clustering using gap-statistic[33]). There was significant correlation with gene expression in all clusters ($P < 0.0001$; $N = 10,000$ bootstraps, range; $0.12 \leq R \leq 0.44$) (Fig. 3c). Each cluster is composed of a large fraction of transcripts where H3K4me3 and gene expression correlate very well (Supplementary Fig. 5). Nonetheless, there are also many transcripts for which H3K4me3 and expression are not correlated. For instance, in cluster 1, many genes have higher gene expression late in MPI than we would predict from H3K4me3 data (far right panels; Supplementary Fig.

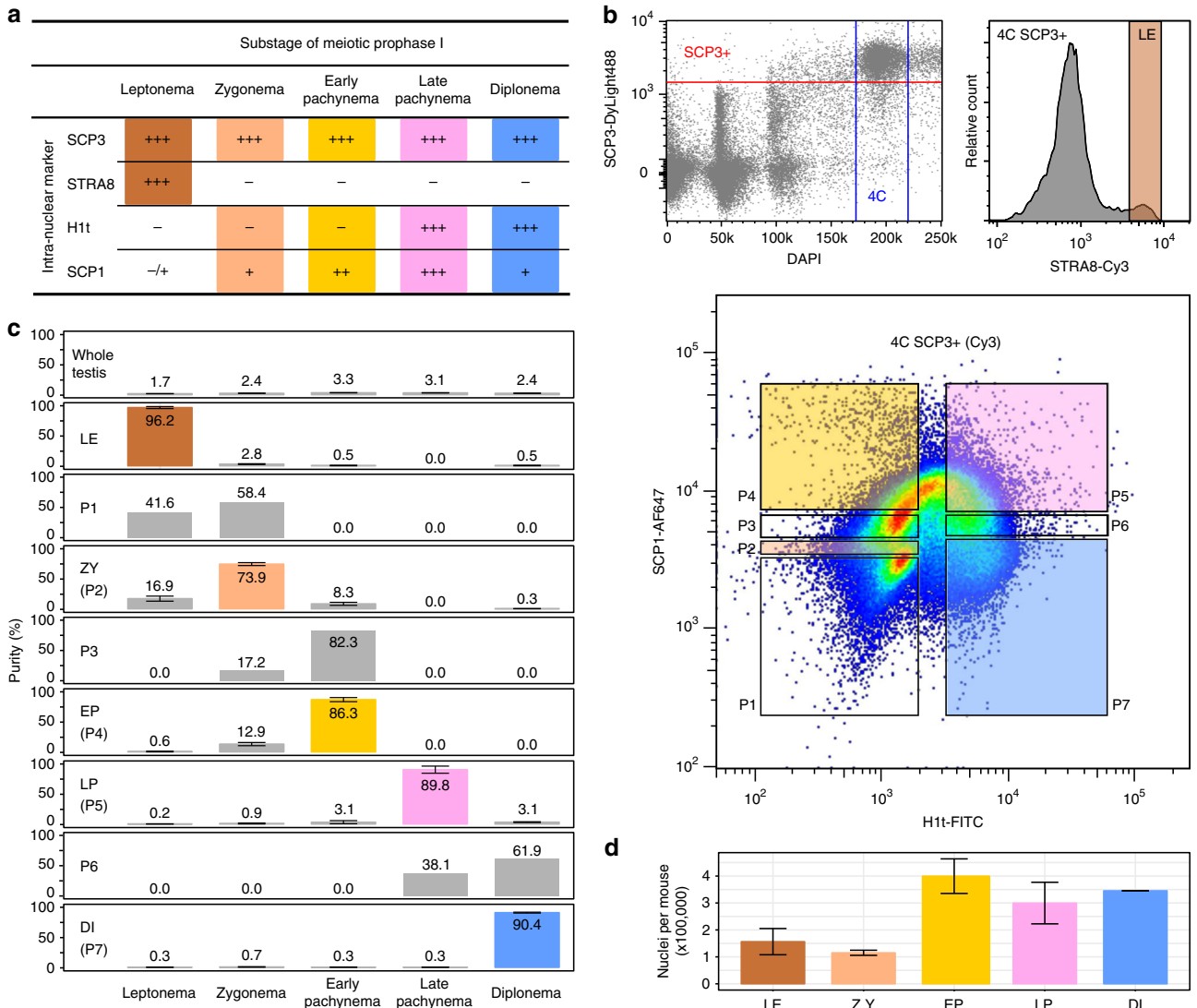

**Fig. 2** Experimental design for isolating stage-specific spermatocyte nuclei. **a** Signal strength of intra-nuclear markers across meiotic substages observed in immunofluorescence staining. Signal strength is classified as absent (−), very weak (−/+), weak (+), medium (++), or strong (+++). Combinatorial markers for isolating stage-specific nuclei are highlighted. **b** Flow cytometric strategies for isolating populations of stage-specific nuclei. Meiotic 4C nuclei are gated by DAPI and SCP3 signals (top left). Stage-specific nuclei are sorted into populations based on combinatorial signals of intra-nuclear markers in two separate sorts; one using antibodies against STRA8 for leptonema (top right), and the other one using antibodies against H1t and SCP1 for seven populations (P1 to P7) (bottom). **c** Distributions and purities of each specific type of nuclei in whole-testis and in sorted populations. The five selected populations of leptonema, zygonema, early pachynema, late pachynema, and diplonema are highlighted as LE, ZY, EP, LP, and DI, respectively. Purities (mean with standard error) of these populations are derived from two or three independent sorts. **d** Numbers of nuclei in each subpopulation collected in an adult mouse. Data (mean with standard error) are derived from two independent sorts. Source data are provided as a Source Data file

5). This may be explained by mRNA accumulation through MPI or by H3K4me3 in early MPI marking poised, but not yet active promoters[34]. In contrast, genes in cluster 5 are less affected by these confounding effects and the correlation is far higher for this cluster. This seems to indicate a concerted wave of gene activation and expression as cells approach the end of MPI. Together, these data demonstrate that ChIP-Seq of H3K4me3 in stage-specific nuclei reveals a dynamic that parallels that of gene expression. These data also reveal an intricate and poorly understood interplay between H3K4me3 and gene expression during MPI.

**Histone modification dynamics at other genomic elements.** Dynamic H3K4-trimethylation is observed at both gene promoters and DSB hotspots in MPI. However, 16% of H3K4me3 peaks (4664/28,528) are outside of these sites (Supplementary Fig.

6a). These unannotated H3K4me3 peaks may represent enhancers, cryptic promoters or other functional elements. To explore these dynamics in greater detail, we used the H3K4me2/3 antibody (Abcam ab8580), as this should capture H3K4me2 modifications that mark enhancers. Indeed, using this antibody, 18,738/53,518 (35%) peaks occur at non-hotspot, non-TSS sites (Supplementary Fig. 7a). Using peaks from this antibody, we derived an MPI profile for each peak across the five MPI stages, then performed unbiased *k*-means clustering. To validate this approach, we examined clustering of DSB hotspots; as expected, hotspots primarily occur in a single large cluster with a maximum signal at zygonema (cluster 3; Supplementary Fig. 7b–d; 89% of hotspots occur in this cluster). Eleven percent of unannotated peaks also occur in this cluster (cluster 3; Supplementary Fig. 7d) and may represent sites of PRDM9 binding that were not detected in hotspot mapping experiments. Alternatively, we cannot rule

**Table 1 Summary of cell-type-specific H3K4me3 ChIP-Seq**

| Substage | Experiment | Antibody | Purity (%) | Starting chromatin DNA (ng) | SPoT (HS; %) | SPoT (All; %) | Peaks (#) |
|---|---|---|---|---|---|---|---|
| Leptonema | 1 | EpiCypher | 96.9 | 2398 | 2.4 | 4.5 | 9528 |
| Leptonema | 2 | EpiCypher | 95.3 | 805 | 2.2 | 7.6 | 13,095 |
| Leptonema | 2 | Abcam | 95.3 | 805 | 3.3 | 20.3 | 30,943 |
| Zygonema | 1 | EpiCypher | 79.7 | 2688 | 4.7 | 6.4 | 15,699 |
| Zygonema | 2 | EpiCypher | 64.6 | 755 | 3.6 | 4.2 | 11,314 |
| Zygonema | 2 | Abcam | 64.6 | 755 | 5.3 | 17.0 | 29,114 |
| Early Pachynema | 1 | EpiCypher | 81.6 | 1793 | 1.6 | 4.2 | 6647 |
| Early Pachynema | 2 | EpiCypher | 89.5 | 4585 | 1.7 | 2.3 | 4883 |
| Early Pachynema | 2 | Abcam | 85.2 | 960 | 2.0 | 14.0 | 17,561 |
| Late Pachynema | 1 | EpiCypher | 84.1 | 4936 | 1.7 | 2.6 | 5189 |
| Late Pachynema | 2 | EpiCypher | 71.3 | 915 | 1.6 | 3.8 | 9124 |
| Late Pachynema | 2 | Abcam | 71.3 | 915 | 1.5 | 20.1 | 19,711 |
| Diplonema | 1 | EpiCypher | 88.7 | 6502 | 1.8 | 3.1 | 6645 |
| Diplonema | 2 | EpiCypher | 82.3 | 1115 | 1.7 | 5.4 | 12,071 |
| Diplonema | 2 | Abcam | 82.3 | 1115 | 1.6 | 21.0 | 18,191 |

out that a cluster of cryptic functional sites simply exhibit a similar dynamic. The remaining unannotated peaks exhibit an MPI dynamic that broadly mirrors that at TSS (Supplementary Fig. 7b, d). It seems likely therefore that functional elements at these sites play a role in regulating meiotic progression. These dynamic, but unannotated sites represent a completely unexplored aspect of the regulation of mammalian meiosis.

Clustering also allowed us to study H3K4me3 dynamics at sites involved in other important processes during MPI. In particular, we examined H3K4me3 sites that are used for DSB targeting in mice that lack functional PRDM9 [27,28]. H3K4me3 peaks used as "default" hotspot locations were predominantly those with maximal signal in early MPI (Supplementary Figs. 6 and 7).

**Identifying other histone marks at hotspots**. At DSB hotspots defined by PRDM9, H3K4me3 is necessary for DSB formation [35]. Nonetheless, H3K4me3 alone is not sufficient to define DSB sites, as this histone mark is also present at other functional sites, such as gene promoters. In mice, H3K36me3 is the only other histone mark described genome-wide at DSB hotspots [17]; however, other histone marks may also be involved in defining DSB sites [36–38]. We therefore used our method to isolate target nuclei from adult mice for a systematic survey of multiple histone modifications.

We performed ChIP-Seq using antibodies against 16 additional histone methylation and acetylation marks in isolated nuclei (Fig. 4a, b). These marks were either previously analyzed at individual hotspots in mice [36,38] or reported to be enriched at DSB sites in other organisms [39,40]. In this exploratory phase, we did not perform ChIP-Seq in all five subpopulations, but instead, in a combined population that includes all stages with potential relevance to understanding histone modifications at DSB sites (leptonema to early pachynema; SCP3+ and H1t− population).

H4K8ac, H4K12ac, H4K20me3, H3K4ac, H3K79me1, H3K79me3, H3K27me1, H3K9me2, H3K9me3, and H3K27me3 were not enriched at hotspots relative to controls (Fig. 4a, b; Supplementary Fig. 8). With the exceptions of H3K27me1, and H3K9me2, these experiments showed enrichment at the expected functional genomic regions (Supplementary Fig. 9), lending confidence the lack of enrichment at hotspots does not result from failed ChIP experiments. Nonetheless, transient or weak signals at hotspots could still be missed.

H3K4me1, H3K27ac, and H4ac5 showed marginal enrichment at hotspots (Fig. 4a, b; Supplementary Fig. 8). The weak signals at hotspots are unlikely the result of suboptimal ChIP-quality, as strong signals are seen at gene promoters and enhancers (Supplementary Fig. 9). Instead, these marks may be rapidly

turned-over or present only in a subpopulation of early spermatocytes. Indeed, a weak DSB-dependent H4ac5 signal was previously reported at two mouse hotspots [36].

PRDM9 can trimethylate both H3K4 and H3K36 [35] and as expected, H3K4me3 and H3K36me3 were both enriched at hotspots (Fig. 4a, b; Supplementary Fig. 8). H3K4me2 and H3K9ac were also strongly enriched at DSB hotspots genome-wide; this clearly resolves the ambiguity from previous studies where these marks were enriched at one but not another hotspot [36]. H3K4me2 is a likely intermediate of PRDM9 H3K4-trimethylation (see above); however, since PRDM9 lacks a histone acetyltransferase domain, it is unlikely that PRDM9 directly acetylates H3K9. In stage-specific experiments, the dynamics of H3K9ac through MPI resemble those of H3K4me3: a signal is first observed at hotspots in leptonema, maximal at zygonema and absent by early pachynema (Fig. 4c). Thus, H3K9ac is a bona fide marker of meiotic DSB hotspots genome-wide, explicitly demonstrating for the first time that proteins other than PRDM9 modify nucleosomes at the sites of DSB hotspots.

H3K4me3 is positively correlated with DSB frequency [28,41], but most variation in hotspot strength is not accounted for by changes in H3K4me3 (Spearman $R^2 = 0.46$) (Supplementary Fig. 10a). H3K36me3 was previously reported to slightly improve the correlation between H3K4me3 and DSB strength [42]; therefore, we tried to better predict hotspot strength using a combination of histone marks that are enriched at DSB hotspots. Multiple linear regression with all histone marks only slightly improved the correlation with hotspot strength (max $R^2 = 0.52$; Supplementary Fig. 10b; see Methods) suggesting that these extra histone marks (including H3K36me3 and H3K9ac) do not offer an independent readout of DSB usage as compared to H3K4me3.

**A histone code distinguishes hotspots from other H3K4me3 sites**. In the absence of PRDM9, DSB hotspots occur at sites of non-PRDM9-mediated H3K4me3 [28]. Nonetheless, in wild-type mice, PRDM9-defined H3K4me3 sites are used. In agreement with previous results [17,18], we found that H3K36me3 is a potent discriminator between DSB hotspots and other H3K4me3-marked sites in the genome (Fig. 4d). Among the other histones, H3K4me2 can best distinguish hotspots (Fig. 4d; Supplementary Fig. 11). We excluded histone marks made by PRDM9 (H3K4me3 and H3K36me3), then performed principal component analysis to explore if any combination(s) of histone marks better define DSB hotspots. The most discriminative PC (PC1) captured signal from H3K9ac and other histone acetylation marks (Fig. 4e), but did not discriminate hotspots as well as H3K4me2

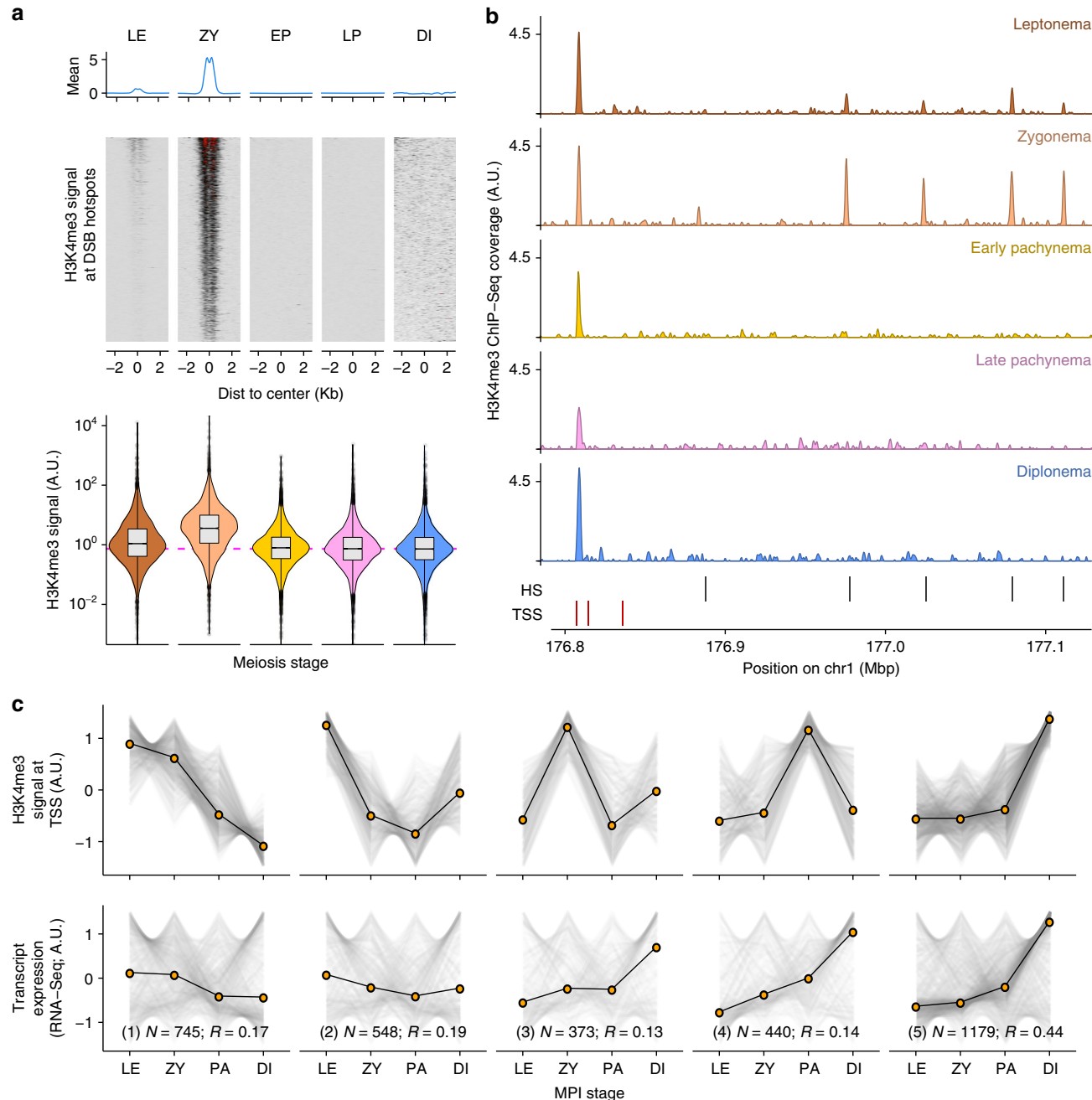

**Fig. 3** H3K4me3 dynamics across meiotic substages. **a** H3K4me3 dynamics at DSB hotspots across five meiotic substages. H3K4me3 signals at DSB hotspots are normalized by the spike-in H4K4me3 mononucleosomes. Quantified H3K4me3 signal is shown in violin plot (lower panel). **b** H3K4me3 dynamics at different genomic elements in five sorted subpopulations of meiotic nuclei. HS hotspots, TSS transcription start sites. **c** Stage-specific H3K4me3 at TSSs recapitulate meiotic gene expression patterns. Genes with only one isoform are selected for this analysis. H3K4me3 profiles at TSSs are grouped into five clusters (top), while the respective mRNA levels are shown in the bottom panel. LE Leptonema, ZY Zygonema, PA Early and Late Pachynema, DI Diplonema. The number of transcripts per cluster (N) and Spearman correlation coefficient (R) for all transcripts are shown. Source data are provided as a Source Data file

or H3K36me3 alone. These data suggest that H3K36me3, H3K4me2 and to a lesser extent H3K9ac and other histone acetylation marks, distinguish the chromatin at sites of PRDM9-marked DSB hotspots from that at other functional sites where H3K4 is trimethylated.

## Discussion

In this study, we developed a method to isolate highly pure populations of meiotic nuclei from whole-testis. This method is rapid, requires very little starting material, and resolves a major hurdle to studying meiosis in mammals. We have demonstrated that we can efficiently sort five populations of MPI nuclei; however, in principle, any number of specific stages can be purified if the requisite antibodies are available. We obtained populations of up to 96% purity for a given meiotic substage, negating the need to artificially synchronize meiosis using chemicals or the need to use juvenile mice to obtain enriched populations of meiocytes. This opens the door to many detailed studies of meiosis and also to experiments in challenging-to-breed knockout mice and

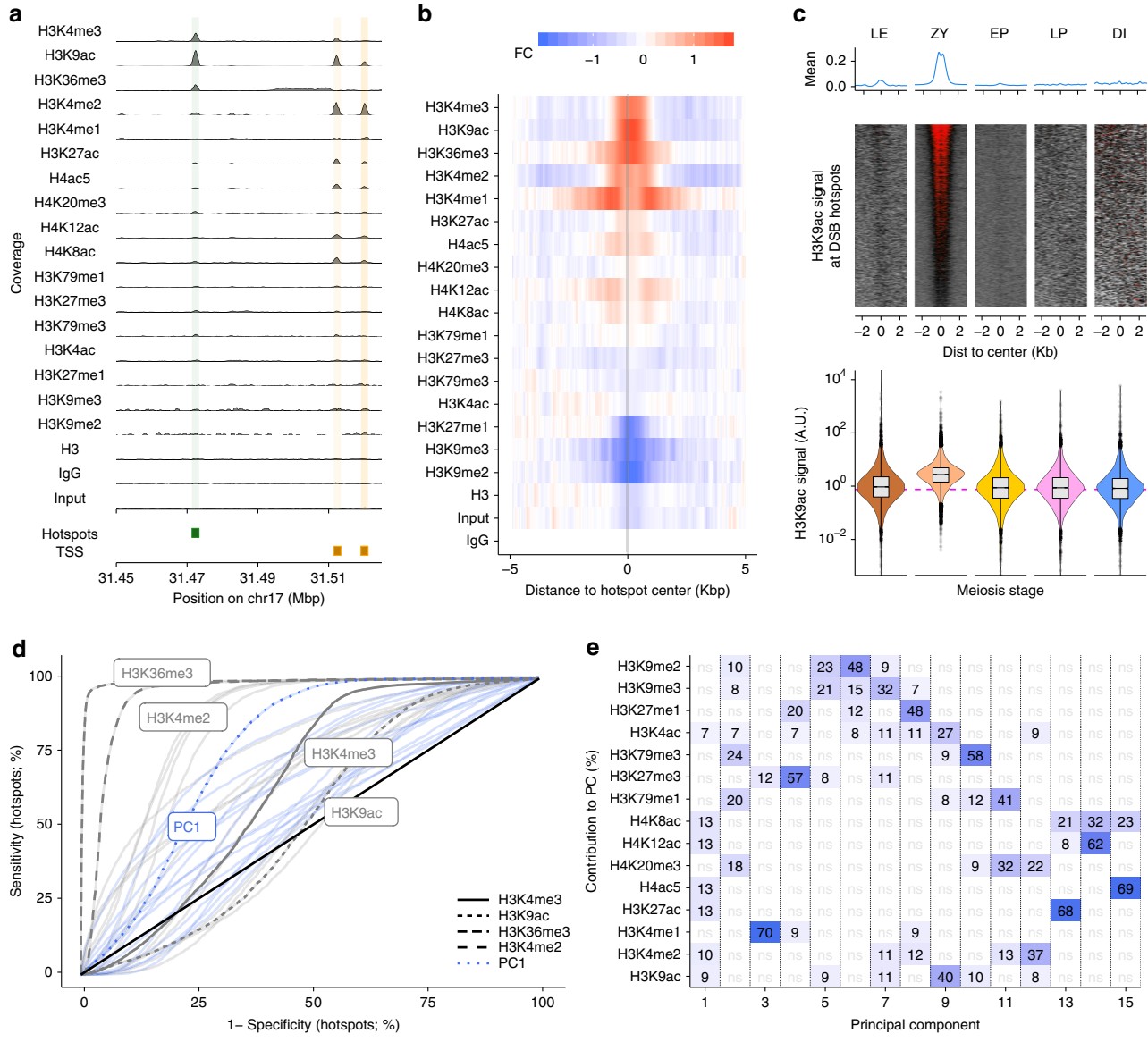

**Fig. 4** Histone modifications at DSB hotspots. **a** Distributions of histone marks along a genomic region. Locations of DSB hotspots are shaded in green, TSS (transcription start sites) in orange. **b** Histone marks are enriched at DSB hotspots. Red depicts enrichment relative to IgG, blue is depletion. **c** H3K9ac dynamics at DSB hotspots across five meiotic substages. Quantified H3K9ac signal is shown in violin plot (lower panel). **d**, **e** Principal component analysis using histone marks can distinguish DSB hotspots from PRDM9-independent H3K4me3 sites. **d** Histone modifications can distinguish DSB hotspots from other H3K4me3 sites. An ROC curve for each histone mark (gray) or PC (blue) is shown. Selected histone modifications and PC1 are highlighted. **e** Contributions of histone marks to each PC. Only marks that contribute more than expected are shown. Deeper blue indicates a stronger contribution (numbers show percent contribution). Data shown in this figure (except **c**) are ChIP-Seq data collected from an early MPI population (leptonema to early pachynema; SCP3+ and H1t− population). Source data are provided as a Source Data file. DSB double-strand break, TSS transcription start sites

importantly in humans, where other synchronization strategies are impractical.

Importantly, fixation prior to sorting allows for genome-wide interrogation of transient protein–DNA interactions in sorted populations. To this end, we profiled H3K4me3 using ChIP-Seq in five MPI populations, capturing extensive dynamics of this histone mark. The H3K4me3 marks made at DSB hotspots by PRDM9 appear in leptonema, are maximal in zygonema, and are removed by early pachynema. Intriguingly, the presence of robust H3K4me3 signal at zygonema implies that PRDM9 trimethylates more nucleosomes than are used during DSB formation. An alternate explanation would require a large fraction of DSBs to form in zygonema. This is unlikely because homolog synapsis, which occurs in zygonema, begins to shut down the DSB

machinery in yeast[43]. An analogous system appears to operate in mammals[44,45]. These are the first data to demonstrate excess H3K4-trimethylation by PRDM9 in wild-type mice; however, recent work in mice coexpressing both endogenous and transgenic PRDM9 reached a similar conclusion[35].

H3K4me3 at gene promoters is also dynamic through MPI. By comparing H3K4me3 at promoters with gene expression, we found that transcript levels are broadly predicted by H3K4me3. The most cohesive group of transcripts exhibit a coordinate increase in H3K4me3 and in expression late in MPI. This likely reflects the resumption of extensive transcription during mid-pachytene and its continued increase into diplotene[46]. A large subset of transcripts exhibit gene expression that appears decoupled from H3K4me3 levels. Some such transcripts exhibit a

strong H3K4me3 signal at the TSS in early MPI, but expression is maximal late in MPI; this may result from accumulation of long-lived mRNA and/or from H3K4me3 marks at the sites of poised RNA polymerase[34]. Indeed, these transcripts are intriguing candidates for further understanding gene regulation in MPI. Because we isolate only nuclei, cytoplasmic mRNA cannot be sequenced in our sorted populations. We therefore used mRNA-Seq data from a previously published study[32]. However, we anticipate that recent developments in transcriptomics using RNA from formaldehyde-fixed nuclei[47] will enable us to adapt our method to study transcription in sorted populations of meiotic nuclei. This may help to clarify the relationship between histone marks and active transcription. Finally, a large fraction of dynamic histone marks (H3K4me2/3) in meiosis do not coincide with gene promoters or DSB hotspots. The functions of these sites are unknown although some are likely to be enhancers. Together, they represent a large, heretofore unstudied set of functional sites that may be modulating mouse meiosis.

To examine the chromatin landscape at DSB hotspots, we also performed the first systematic genome-wide survey on a wide repertoire of histone marks. Our ability to efficiently sort early MPI nuclei eliminated the need to perform these experiments in juvenile mice; this negates the logistical challenge of collecting sufficient juvenile animals and assures consistent and high levels of enrichment. Aside from H3K4me3[28] and H3K36me3[17,18], H3K9ac, H3K4me2, H3K4me1, H3K27ac and H4ac5 were enriched at DSB hotspots genome-wide. Incorporating the extra information about these histone marks only marginally increases our ability to predict hotspot strength, implying that even the expanded chromatin landscape at hotspots remains a poor predictor of DSB formation. The genome is replete with H3K4me3 marks that are used for DSB formation in the absence of PRDM9; however, enigmatically, these sites are not used if PRDM9 is functional. We demonstrated that both H3K36me3[17,40,42], and multiple histone marks distinguish between these sites. Although this unique histone "code" may help the DSB machinery to distinguish PRDM9-defined H3K4me3 from other H3K4me3 marks, it is equally possible that direct interactions mediated by PRDM9 itself are important for this distinction. It seems unlikely that PRDM9 is responsible for all of the histone tail modifications at hotspots; therefore, other histone remodelers are likely acting at the sites of PRDM9-marked H3K4/K36me3. H3K9ac and H3K4me3 are strength correlated and have a similar temporal dynamic implying that they occur at a comparable frequency at hotspots. In mitotic cells, H3K9 acetylation is actively promoted at H3K4 trimethylated nucleosomes[48]. It is therefore likely that H3K9ac is a constitutive response to H3K4 trimethylation. Nonetheless, histone acetylation may be functionally important per se in the context of DSB formation; for example, in fission yeast, H3K9 to H3A9 mutation eliminates H3K9ac and consequently reduces DSB formation[40].

Overall, we demonstrated that our straightforward and robust method can explore previously hidden dynamics of mammalian meiotic recombination. Importantly, our method is widely applicable to other organisms, tissues and cell types, paving the way for further understanding of the temporal dynamics of other developmental processes.

## Methods
**Mouse strains**. C57BL/6J (B6) mice were either obtained from The Jackson Laboratory (Stock no. 000664) or bred in-house. All experiments were done on adult mice (≥8 weeks of age). Mice were sacrificed in accordance with the NIH Animal Care and Use regulations.

**Ethical compliance**. The authors confirm that all experiments were performed in accordance with the NIH Animal Care and Use regulations.

**Antibodies**. The following antibodies were used for primary immunofluorescence staining: anti-SCP3 (D-1) (Santa Cruz, sc-74569), anti-STRA8 (Abcam, ab49602), anti-H1t (a gift from Mary Ann Handel and a custom-made antibody), and anti-SCP1 [Biotin] (Novus Biologicals, NB300-229B).

The following antibodies were used for secondary immunofluorescence staining: goat anti-mouse IgG conjugated with Cy3 (Jackson ImmunoResearch, 115-167-003), goat anti-rabbit IgG conjugated with FITC (Jackson ImmunoResearch, 115-097-003), goat anti-mouse IgG conjugated with DyLight488 (Abcam, ab98757), goat anti-rabbit IgG conjugated with Cy3 (Jackson ImmunoResearch, 111-165-003), goat anti-guinea pig IgG conjugated with Alexa Fluor 488 (Thermo Fisher Scientific, A-11073), streptavidin conjugated with Alexa Fluor 647 (Thermo Fisher Scientific, S21374).

The following antibodies were used for ChIP: anti-H3K4me3 (Abcam, ab8580), anti-H3K4me3 (EpiCypher, 13-0028), anti-H3K9ac (Active Motif, 39918), anti-H3K4me2 (Active Motif, 39914), anti-H3K36me3 (Active Motif, 61102), anti-H4ac5 (Millipore, 06-946), anti-H3K4me1 (Abcam, ab8895), anti-H3K27ac (Abcam, ab177178), anti-H3K4me2 (Abcam, ab177178), anti-H4K8ac (Abcam, ab15823), anti-H4K12ac (Active Motif, 39928), anti-H3K27me3 (Millipore, 07-449), anti-H4K20me3 (Millipore, 07-463), anti-H3K79me1 (Abcam, ab2886), anti-H3K4ac (Abcam, ab176799), anti-H3K79me3 (Abcam, ab2621), anti-H3K9me2 (Abcam, ab1220), anti-H3K9me3 (Active Motif, 39766), anti-H3K27me1 (Millipore, 07-448), anti-H3 (Abcam, ab1791), normal rabbit IgG (Millipore, 12-370).

**Nuclei preparation**. Testes from adult mice were de-capsulated and fixed with 1% formaldehyde for 10 min followed by 5 min of quenching at RT. Fixed tissues were homogenized and filtered through a 70- or 100-μm cell strainer. After washing with chilled 1× PBS (phosphate buffered saline), nuclei were extracted using nucleus extraction buffer (15 mM Tris-HCl pH 7.4, 0.34 M sucrose, 15 mM NaCl, 60 mM KCl, 0.2 mM EDTA (ethylene diamine tetraacetic acid), 0.2 mM EGTA (ethylene glycol tetraacetic acid)) on ice for 5 min and were homogenized with 20 strokes with loose pestle followed by 10 strokes with tight pestle. Nuclei were filtered through a 40-μm cell strainer and resuspended in chilled PBTB buffer (1× PBS with 0.1% Triton X-100, 5% bovine serum albumin and protease inhibitor).

**Immunofluorescence staining**. Nuclei were incubated in 10% of normal serum at RT for 10 min. This blocking step helps reduce nonspecific binding of antibodies. Nuclei were then labeled with different combinations of primary antibodies (1 μg of antibodies to 11 million events) in 10% of normal serum at 20 °C for 40 min. Nuclei were washed, resuspended in PBTB, blocked with serum, and then labeled with secondary antibodies (1:250 dilution) in 10% of normal serum at 20 °C for 30 min. Controls for each secondary antibody were also prepared for setting up the threshold of background fluorescent signal for FACS. Nuclei were washed, resuspended in PBTB, and stored at 4 °C until sorting. A potential concern is that antibodies from the sorting step may be detected by ChIP. Such an effect was not seen in our experiments (see No-antibody control in Supplementary Table 1); however, care should be taken in the choice of antibodies to minimize this possibility.

**Counting stage-specific nuclei from whole-testis**. Nuclei isolated from whole-testis were stained with antibodies against SCP3, H1t and SCP1, and subsequently with secondary antibodies as described. Immunofluorescence-labeled nuclei were spread on microslides, stained with mounting medium with DAPI (Vetashield), sealed with coverslips, and viewed using fluorescence microscopy. Each of the five types of MPI nuclei (SCP3+) were counted; these were characterized by the patterning of the SCP3 and H1t proteins (Fig. 1). A total of 678 nuclei were counted from two independent experiments. To extrapolate counts to the percentage of cells of each stage in whole-testis (top panel in Fig. 2c), we use the following formula:

$$WT_{stage}(\%) = N_{stage}/678 \times (N_{4C}/N_{all}) \times 100,$$

$N_{stage}$ = number of nuclei of each stage;

$N_{4C}$ = number of total 4C (SCP3+) nuclei from sorting;

$N_{all}$ = number of total singlet nuclei (1C to 4C) from sorting.

**Quantification of immunofluorescence signal intensity**. Nuclei isolated from whole-testis were stained with antibodies against SCP3 and STRA8 or with antibodies against SCP3, H1t and SCP1, subsequently with secondary antibodies, and spread as previously described. Immunofluorescence signals of each marker were quantified using Volocity 6.2.1 (PerkinElmer). 4C nuclei were identified using signals from the DAPI channel by excluding objects with size <40 μm². Touching objects were separated using the parameter of 30–50 μm². Stage-specific nuclei in MPI were characterized by the presence of SCP3 signal and by the patterning of synaptonemal complex[14]. Signals from all channels were recorded and exported for signal quantification.

**Isolation of meiotic subpopulations with FACS**. Nuclei were filtered through a 40-μm cell strainer, and stained with DAPI for 30 min or longer at RT before sorting. All sorting experiments were performed on either a BD FACSAria II or a

BD FACSAria Fusion flow cytometer at a flow rate of ~20,000 events/s. Singlets were gated using both forward scatter and side scatter. Nuclei from primary spermatocytes (4C) were gated based on DNA content deduced from the DAPI signal, and sorted into populations of interest based on fluorochrome intensity into collection tubes containing PBTB. Sorted nuclei were collected by centrifugation. Nuclei were examined under microscope and counted with a hemocytometer. Finally, excess buffer was removed and nuclei pellets were stored at −80 °C.

**Assessment of purity of sorted populations**. An aliquot of nuclei suspension from each sorted population was spread on microslides, further stained with mounting medium with DAPI (Vectashield), and sealed with coverslips for purity examination. About 150 nuclei from each sorted population were checked at ×400 magnification using immunofluorescence microscopy. The staging was primarily accessed by the patterning of SCP3 [14] and further confirmed by signals of STRA8 or of H1t and SCP1 accordingly.

**Chromatin extraction**. Chromatin extracted for ChIP-Seq was either sheared by sonication or Micrococcal nuclease (MNase) digestion. Chromatin for all H3K4me3/H3K9ac-ChIP-Seq for studying histone dynamics in stage-specific populations was sheared by sonication, whereas ChIP-Seq for identifying histone marks at hotspots were performed on MNase-digested chromatin. Experimental procedures are listed below in detail.

**Shearing chromatin by sonication**. Frozen nuclei pellets were thawed at RT for 10 min. Nuclei were lysed with Lysis buffer 1 (0.25% Triton X-100, 10 mM EDTA, 0.5 mM EGTA, 10 mM Tris-HCl pH 8) and incubated at RT for 10 min. Nuclei pellets were subsequently washed with Lysis buffer 2 (200 mM NaCl, 1 mM EDTA, 0.5 mM EGTA, 10 mM Tris-HCl pH 8) and lysed with RIPA buffer (10 mM Tris-HCl pH 8, 1 mM EDTA, 0.5 mM EGTA, 1% Triton X-100, 0.1% sodium deoxycholate, 0.1% SDS (sodium dodecyl sulfate), 140 mM NaCl plus protease inhibitor). Chromatin was sheared into ~50–300 bp fragments by sonication using Bioruptor (diagenode). The amount of chromatin DNA was measured using a Qubit dsDNA HS Assay Kit (Thermo Fisher).

**MNase digestion**. Frozen nuclei pellets were thawed at RT for 10 min. Nuclei pellets were resuspended with MNase buffer (50 mM Tris-HCl pH 8, 1 mM CaCl$_2$, 4 mM MgCl$_2$, 4% NP-40 plus protease inhibitor). MNase digestion was performed in a concentration of 3U MNase (USB, Affymetrix) per one million nuclei at 37 °C for 5 min. The reaction was stopped by adding a final concentration of 10 mM EDTA and incubated at 4 °C for 5 min. Soluble chromatin was collected and diluted with RIPA buffer.

**Chromatin immunoprecipitation**. Chromatin was immunoprecipitated with 0.8–5 μg antibodies (or 5 μl unpurified serum) in 0.5–1 ml RIPA buffer at 4 °C overnight (see Supplementary Table 1 for details). For stage-specific H3K4me3-ChIP-Seq, modified mononucleosomes (SNAP-ChIP K-MetStat Panel, EpiCypher) were added at a spike-in volume of 1 μl per ~1.5 μg chromatin DNA before ChIP. The immuno-complexes were captured using 20–75 μl Dynabeads Protein G (30 mg/ml, Novex) at 4 °C for 2 h. The beads were washed once with low salt buffer (0.1% SDS, 1% Triton-X-100, 2 mM EDTA, 20 mM Tris-HCl pH 8, 150 mM NaCl), once with high salt buffer (0.1% SDS, 1% Triton-X-100, 2 mM EDTA, 20 mM Tris-HCl pH 8, 500 mM NaCl), and twice with LiCl buffer (0.25 M LiCl, 1% IGEPAL-CA630, 1% sodium deoxycholate, 1 mM EDTA, 10 mM Tris-HCl pH 8). ChIPed DNA was eluted using a IPure kit v2 (diagenode).

**Sequencing library construction**. ChIP-seq libraries were constructed with a KAPA Hyper Prep Kit (Kapa Biosystems) following steps for generating 1 μg of library DNA. DNA libraries were cleaned up with an Agencourt AMPure XP system (Beckman Coulter). The DNA concentration and fragment size of these libraries were measured with a Qubit dsDNA HS Assay Kit (Thermo Fisher) and an Agilent High Sensitivity DNA Kit (Agilent), respectively.

**High-throughput DNA sequencing**. DNA sequencing was performed on the Illumina HiSeq 2500 or HiSeq X. Sequencing tags were aligned to the mouse mm10 reference genome using BWA mem 0.7.12 [49]. For ChIP-Seq samples containing the K-MetStat spike-in panel, we aligned to a custom mm10 genome to which we added the sequences of the DNA bound by each K-MetStat nucleosome.

**Normalization and peak strength estimation using spike-in**. We retained only sequencing reads with an alignment q-score ≥20, that were mapped in a "proper pair" (for paired-end data), that passed vendor QC and that mapped as a primary alignment. For paired-end data, we only retained reads from fragments with an insert size between 100 and 200 bp. Normalization using the K-MetStat spike-in was performed as described previously [29], with slight modification. Briefly, each spike-in nucleosome is bound to a unique DNA sequence. Thus, the number of reads recovered for each nucleosome is a proxy for the frequency of that nucleosome in the library. We first converted paired-end reads to fragments, then

calculated the enrichment of fragments mapping to each spike-in sequence after ChIP:

$E_{si}$ = (barcode fragments in ChIP/barcode fragments in input).

Next, we calculated H3K4me3 ChIP-Seq fragment coverage in all 25 bp nonoverlapping windows in the genome. The per-locus enrichment was then calculated as:

$E_{locus}$ = (fragment coverage in ChIP)/(fragment coverage in input).

The histone modification density was then calculated as:

$$\text{HMD}(\%) = 100 \times (E_{locus}/E_{H3K4me3\_A}).$$

$E_{H3K4me3\_A}$ is the $E_{si}$ value for the H3K4me3_A spike_in. For our final estimate of H3K4me3 signal, we normalized the HMD by the global average HMD. The ChIP-Seq signal at peaks and other genomic intervals was calculated as the sum of the signal in all overlapping 25 bp windows.

**Normalization and peak strength estimation using promoters**. The K-MetStat panel cannot be used to normalize data from H3K9ac ChIP-Seq. Thus, we used an alternative strategy using stable promoters. NCIS [50] was used to estimate the contribution of background to each sequencing library. We then calculated the NCIS normalized ChIP-Seq fragment coverage in all 25 bp windows in the genome ($c$ = ChIP coverage − (Input coverage × NCIS factor)). The ChIP-Seq signal at the central ±1 Kbp region around GENCODE transcript 5′ ends (TSS) was quantified as the sum of the coverage in all 25 bp widows overlapping the site. TSS coverage was estimated in each of the five meiotic populations. TSS that overlapped a DSB hotspot, TSSs with a negative strength after correction and TSS with the strongest (1%) and weakest (15%) H3K4me3 signal at each stage were discarded. The log$_2$ ratio of NCIS-corrected H3K4me3 was calculated at each TSS between all pairs of stages. TSSs with an absolute log$_2$ (fold-change) ≤ 1.8 between all stages were retained. This resulted in 362 TSSs for H3K4me3 and 187 TSSs for H3K9ac. The median signal across these intervals was calculated for each ChIP and used as a normalization factor. The strength of each 25 bp genome interval was calculated as the NCIS-corrected coverage divided by the normalization factor. The ChIP-Seq signal at peaks and other genomic intervals was calculated as the sum of the signal in all overlapping 25 bp windows. Note that for H3K4me3 ChIP-Seq, this internal normalization gives comparable results to those using the K-MetStat spike-in panel (Supplementary Fig. 4).

**Correlating H3K4me3 profiles at TSSs with mRNA expressions**. We used RNA-Seq data from spermatocyte populations obtained by classical cell sorting to quantify gene expression [32]. Gene expression at GENCODE vM20 transcripts was quantified for each RNA-Seq dataset using kallisto 0.45.0 (kallisto quant --single --single --single-overhang --fragment-length = 250 --sd = 100 --seed = 42). Only transcripts with maximal expression >0.2 tags per million (TPM), with at least a 1.25-fold change in expression and with nonzero expression and H3K4me3 signals at all stages were considered for subsequent analysis. To minimize the confounding effects of multiple isoforms, only TSS with a single transcript in this list were retained.

The temporal expression profile for each transcript was described as a four unit vector (Leptonema, Zygonema, Pachynema, Diplonema). Similarly, a temporal profile of H3K4me3 at the TSS was generated. We averaged the Early Pachynema and Late Pachynema H3K4me3 signals to allow direct comparison with the RNA-Seq data.

The overall correlation between H3K4me3 and gene expression was calculated by concatenating all gene expression vectors and comparing with a concatenated vector of H3K4me3. To calculate the expected random correlation, we shuffled the order of each 4-unit transcript H3K4me3 vector before concatenation. In all, 10,000 iterations of this process were performed.

$k$-means clustering was used to group similar H3K4me3 temporal profiles and to compare each set to gene expression. We used an implementation of the gap-statistic to determine the optimal number of clusters (R; factoextra package). Within each set of clusters the method described in the previous paragraph was employed to determine the correlation between H3K4me3 and expression temporal profiles and to derive an empirical $p$ value for this correlation. Clustering by gene expression instead of by H3K4me3 signal gives similar correlations, though the number of clusters can change (data not shown).

**Peak calling**. Peaks for H3K4me3 ChIP-Seq were called using MACS2 (version 2.1.2) [51] with default parameters except (-q 0.1 --broad) and with a stage-matched input DNA sample as a control. Peaks overlapping DSB hotspots or gene promoters were ascertained using bedtools (version v2.27.1) [52] after removing black-listed regions described in ref. [53]. DSB hotspots were defined and reported in previous studies [28]. TSSs (TSSs) were defined as the ±0.5 Kbp region around GENCODE v20 transcripts [54].

**Unbiased clustering**. ChIP-Seq peaks from all five populations were merged and peak strength was calculated (as described above) for each merged peak in each population. The H3K4me3 profile for each peak was described as a five unit vector (LE, ZY, EP, LP, DI) and scaled by subtracting the mean and dividing by the

standard deviation. k-means clustering was used to cluster. Five clusters was determined as optimal using the gap-statistic.

**Multiple linear regression**. Hotspots overlapping a TSS or overlapping a site used for DSB formation in $Prdm9^{-/-}$ mice were discarded. Only hotspots with read coverage >0 for all histone marks were used for regression analyses because we performed regression on the log10 transformed coverage values. The leaps package in R was used to perform an all-subsets regression using the seven histone marks enriched at DSB hotspots.

**Principal component analysis**. All H3K4me3 peaks from LE, ZY, EP and LP were used for analysis. Each interval was resized to ±250 bp around the center. Sequencing reads for each histone mark were counted at each feature. Input DNA reads were also counted and subtracted from the count for each mark, following NCIS[50] correction. DSB hotspots and GENCODE TSSs were expanded to ±1500 bp to determine H3K4me3 peaks that overlapped hotspots and TSSs, respectively. Peaks that overlapped both a hotspot and TSS were discarded as the potential compound signal would confound these analyses. Only autosomal peaks were used. H3K36me3 and H3K4me3 were excluded. The R prcomp command was used for Principal Component Analysis. Variables were scaled to have unit variance and zero centered.

ROC curves were built by ranking intervals by the dependent variable. Intervals were ranked either from high to low or from low to high and the ROC with the higher area under the curve was used.

**Reporting summary**. Further information on research design is available in the Nature Research Reporting Summary linked to this article.

## Data availability
The authors declare that all data generated or analyzed during this study are included in this published article (and its supplementary information files). The sequencing data reported in this paper are archived at the Gene Expression Omnibus (www.ncbi.nlm.nih.gov/geo) as accession no. GSE121760. The source data for all figures are provided as a Source Data file. All data are available from the authors on reasonable request.

## Code availability
The custom pipeline used for these analyses is deposited at zenodo (https://doi.org/10.5281/zenodo.2651204)[55].

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

## Acknowledgements

We thank members from Camerini-Otero's lab for discussions, and Galina Petukhova for insightful comments. We are grateful to Mary Ann Handel for sharing H1t antibodies. This study used the high-performance computational capabilities of the Biowulf Linux cluster at the National Institutes of Health, Bethesda, MD (http://biowulf.nih.gov). We thank members from the NHLBI Flow Cytometry Core for assistance with nuclei sorting, and members from the NIDDK Genomics Core for assistance with high-throughput sequencing. This work was funded by the NIDDK Intramural Research Program (R.D.C.-O.).

## Author contributions

K.-W.G.L., K.B., G.C., F.P., and R.D.C.-O. conceived the study and designed the experiments. K.-W.G.L. performed the experiments. K.-W.G.L., K.B. and F.P. analyzed the data. K.-W.G.L., K.B., F.P. and R.D.C.-O. wrote the manuscript.

## Additional information

**Competing interests:** The authors declare no competing interests.

