## [Peer Review File · Nature Communications]

Reviewers' comments:

Reviewer #1 (Remarks to the Author):

Lam et al report a method for the isolation of nuclei from all substages of meiotic prophase I male germ cells of mice. The authors demonstrate that the material isolated in this manner is suitable for ChIPseq studies and describe the dynamics of several histone modifications in the course of meiotic prophase. Overall, this study is of high quality and will be appreciated by meiosis, germ cell, and epigenetic fields. My only gripe is with the fact that Abstract does not reflect correctly on the substance of the method. It states that the paper presents "a simple method for isolating pure subpopulations of meiocytes". However, while the simplicity of the method is subjective, the statement is incorrect since the method isolates nuclei rather than cells. Therefore, Abstract should clearly state that the method isolates nuclei only while Introduction (and where appropriate elsewhere in the manuscript) should say that the method does not allow gene expression studies and necessitates either additional purification of germ cells for RNAseq or availability of preexisting data obtained by classical sorting methods.

Reviewer #2 (Remarks to the Author):

In this manuscript, the authors present a new method for retrieving highly specific population of male germ cells in the substages of meiosis I prophase. They show the utility of this with an extensive ChIP-seq analysis profiling histone modifications throughout meiotic prophase. The findings enlarge our understanding of the much-studied histone H3K4me3 marks at both genetic recombination sites and promoters, and also reveal considerable complexity throughout meiotic prophase of other, less well-studied histone marks. Most importantly, this work provides a method that can be exploited for many other studies of dynamic transitions during meiotic prophase.

Premise: For decades, researchers in the field of mammalian meiosis have attempted to refine and scale methods for the study of chromatin and cellular dynamics during meiotic prophase, especially in the male, where its study is experimentally more tractable than in the female. Prophase substages are defined cytologically and are a continuum rather than abrupt or static stages, a fact well reflected in gene expression data. There are no marker proteins unique to single substages, although several recent studies have successfully exploited combinatorial arrays of markers. Moreover, although spermatocytes increase in cell size, size differences are an imprecise means for sorting the substages. These features define the challenges to temporal mapping of meiotic prophase events that were tackled by the authors, and are set up well in the introduction.

Methods: Key to the nuclear FACS strategy was a set of reliable intra-nuclear protein antibodies, no one of which was unique to a single substage, but which in combinatorial arrays could discriminate the sub-stages of meiotic prophase. These were standard markers that have been used in other studies to define substages, but this is seemingly the first time that they have been in a strategy to sort nuclei. The results were impressive: reasonable purity and the cells look at least somewhat like what they are supposed to be by classical cytological criteria (Suppl. Fig. 1). The modifier "somewhat" is used because the cells are not optimally spread; most likely the prior fixation renders these preparations more similar in resolution and detail to whole mounts than to spread chromatin, a caveat that the authors might want to add. In view of the objectives of the present study, this is a minor concern and in any case, probably not resolvable at this time; nonetheless it does pose a cautionary note that these preparative procedures, with initial fixation (crosslinking), may not be suitable for all studies of meiotic chromosome dynamics. Encouragingly however, cells were obtained in sufficient number for standard genomic protocols.

Results: The meiotic prophase sub-stage-specific cell populations were subjected to ChIP-seq to detect a number of histone modifications, with interesting, but not entirely conclusive, results and insights. For example, H3K4me3 marks at recombination hotspots were confirmed but temporal

dynamics reveal that there is more to be learned about the removal of these marks as recombination proceeds. For the most part, H3K4me3 marks at TSS correlated with RNA-seq data on transcription (even though the latter data were not as highly resolved temporally as the present analysis on histone marks). Perhaps most interesting with respect to H3K4me3 marks is that a substantial proportion of them occur at unannotated sites, clearly a world yet to discover. Analysis of other histone marks at hotspots makes steps towards defining a histone code unique to hotspots. Of particular interest is the finding of a strong H3K9ac signal, demonstrating there may be more to this story than PRDM9. However, in spite of an exhaustive analysis, no definitive histone code was found to be clearly unique to or predict strength of hotspots. Indeed, it is known that there are more genomic PRDM9 sites activated than DSB sites, and more DSBs than there are reciprocal recombination events. Thus the epigenetic modifications responsible for this progressive selection are of major importance. This study points a way to resolve this issue.

Significance: In brief summary, both the method and the findings on histone modifications are of value to the community. The significance and impact of this paper will be considerable, and, in the authors' words, these studies "open the door" to avenues for better understanding of the molecular dynamics of chromatin in meiotic prophase I.

Reviewer #3 (Remarks to the Author):

NCOMMS-18-34243-T

Lam et al., "Cell-type-specific genomics reveals histone modification dynamics in mammalian meiosis"

<General comments>

In this study, authors developed a new method to purify five specific population of meiotic cells from adult mouse testis by a relatively simple FACS sorting. Although this method seems to be limited to fixed nuclei thus cannot be utilized for RNA and native protein isolations, this long-awaited method has a potential impact on the study of meiosis especially using mammalian species. Authors demonstrated changes in cell-type specific histone modification during MPI, some of which are associated with hotspots and gene promoters. On the other hand, unfortunately, the biological findings demonstrated by the authors are still within the scope of previous findings and lack the novelty. In addition, I also have several concerns regarding both technical and writing aspects described as follows;

<Major comments>

1) "Introduction" was poorly described; there was little information about the history of this research field and what previous studies have done. Instead, there were lots of description about the results and discussion of this study, so it's just like an another "abstract".

2) For data normalization between different samples, authors utilized a method called "NCIS" proposed by Liang et al. in 2008. This method is well-known, but not popular in the field. Although authors seemed to normalize the inter-sample variations carefully and manually, I'm not still certain that their normalization was appropriate, especially because they observed the most prominent H3K4me3 peaks at hotspots in zygotene, while PRDM9 reaches the highest level in leptotene according to the previous studies. In Fig. 3B, zygotene exhibited a noisy background, while others didn't have such backgrounds. Does this affect the peak intensities in zygotene? Whatever the reason, I highly recommend to perform "Spike-in" for more precise normalization, particularly because this kind of study will be recognized as a standard, and many researchers in the field will possibly refer in the future.

3) The authors made some discussion about the balance among DSBs, H3K4me3 and PRDM9 in the "Result" section (page 8, line 189-196) without showing data. Because this point is very

important and substantial data are essential, I suggest to employ Prdm9 KO cells to verify their thought.

4) In the paragraph of page 8 line 202-224, the description sometimes unmatched to the Fig. 3D. First of all, authors described that "We found that the H3K4me3 profiles at TSSs were positively correlated with gene expression through MPI (Fig. 3D, $R = 0.32$; Spearman test)", while I do not see such data in the figure. Second, authors mentioned that "This is particularly evident in cluster 1, where many genes have higher mRNA levels late in MPI than we would predict from H3K4me3 data (Supplementary Fig. 3). This may be explained by mRNA accumulation through MPI, or by H3K4me3 marking poised, but not yet active promoters". How do authors think that way, as there are decreased levels of H3K4me3 and significantly higher levels of transcripts in the late MPI? Furthermore, it would be interesting to see how many of H3K4me3 marks on the gene promoters are regulated by PRMD9. If they are not made by PRDM9, is there any candidate methyltransferase, such as SET1? In addition, are these H3K4me3 marks on gene promoter specific to meiocytes? How about in spermatogonia?

5) In the paragraph of page 9 line 226-251, authors explained and "discussed" about Fig. S4. Although readers may be able to assume, there was no indication of cluster names in the figure, and it's difficult for me to understand their explanation based on the exhibiting dataset due to their rough and unkind description. Also in the same paragraph, authors claimed that these PRDM9-dependent H3K4 methylation varied during MPI and that many H3K4 methylation retained until zygotene. However, this is not surprising because only 200-300 DSBs can be formed during MPI. It would be more interesting to see if these H3K4 methylation marks are generated in a spatiotemporal manner. Use of Spo11-KO would be helpful.

6) In Fig. 4, S5, and S6, authors performed ChIP-seq using 17 antibodies of modified histones. I highly appreciate their effort, while according to the Fig. 3a and Table S1, the cell number for the ChIP-seq (16,000) seems insufficient for global peak detection (note that Fig. 3a was calculated for H3K4me3, and other antibodies have much less ChIP efficiency thus they may require more number of cells).

7) Although it's not directly linked to this study, comparison of meiotic cell properties between juvenile and adult appears to be interesting, as the first and subsequent round of spermatogenesis can be different, while no substantial studies have demonstrated this question.

<Minor comments>

8) As also pointed out above, lots of speculation and discussion were included in the result section. This is very confusing.

9) Supplementary Fig. 2c was referred in the manuscript (page 8, line 190), but the figure is missing.

Response to reviewers' comments on manuscript NCOMMS-18-34243-T:

General response:

In revising the manuscript, we have made substantial changes. These have arisen from the request from reviewer #3 to perform “spike-in” experiments to quantify H3K4me3 signal at each stage of meiotic prophase I (MPI). These experiments have not changed the major conclusions of the paper but did necessitate some structural changes that were not requested by referees.

We performed new ChIP-Seq experiments into which we spiked in a panel of H3 methylated mononucleosomes prior to chromatin immunoprecipitation. These spike-in experiments allowed us to determine the ChIP efficiency in each sample, and to normalize accordingly. Because we spiked in a panel of modified nucleosomes, we could also assess antibody cross-reactivity and intriguingly, we found that the H3K4me3 antibody (Abcam, ab8580; the most commonly used H3K4me3 antibody in the field), also recognizes H3K4me2 (see also, Shah et al., Mol. Cell 2018). We therefore repeated the spike-in experiment for all five MPI populations in duplicate with another H3K4me3 antibody (EpiCypher, 13-0028) that was highly specific to H3K4me3. Despite using another H3K4me3 antibody and spike-in normalization, the dynamics of H3K4me3 at hotspots remained similar to those described in the original submission.

One difference, unrelated to the choice of antibody, is that in our new experiments, H3K4me3 is not present at hotspots in early pachynema. This was confirmed in replicate experiments in the five MPI populations with the specific H3K4me3 antibody, in a new experiment using the non-specific H3K4me3 antibody, and in a new experiment using H3K9ac antibody. This difference arose because a less stringent threshold was used to split zygotene and early pachytene nuclei in our original submission. Thus, hotspot signal from some zygotene nuclei was detected in early pachytene population. Because of this, we have now included the purity of each sorted population used for the new H3K4me3-ChIP-Seq in Table 1.

Note that the use of new data has slightly changed downstream figures and analyses, but the core conclusions remain as per the original submission.

Summary of additional experiments:

H3K4me3 ChIP-Seq in five MPI populations (replicate I)

H3K4me3 ChIP-Seq in five MPI populations (replicate II)

H3K9ac ChIP-Seq in five MPI populations

H3K4me2/3 ChIP-Seq in five MPI populations

Input DNA libraries in five MPI populations (replicate I)

Input DNA libraries in five MPI populations (replicate II)

Reviewer #1 (Remarks to the Author):

We would like to thank the reviewer for their comments and suggestions.

Lam et al report a method for the isolation of nuclei from all substages of meiotic prophase I male germ cells of mice. The authors demonstrate that the material isolated in this manner is suitable for ChIP-seq studies and describe the dynamics of several histone modifications in the course of meiotic prophase. Overall, this study is of high quality and will be appreciated by meiosis, germ cell, and epigenetic fields.

My only gripe is with the fact that Abstract does not reflect correctly on the substance of the method. It states that the paper presents “a simple method for isolating pure subpopulations of meiocytes”. However, while the simplicity of the method is subjective, the statement is incorrect since the method isolates nuclei rather than cells. Therefore, Abstract should clearly state that the method isolates nuclei only while Introduction (and where appropriate elsewhere in the manuscript) should say that the method does not allow gene expression studies and necessitates either additional purification of germ cells for RNAseq or availability of preexisting data obtained by classical sorting methods.

R1.1: We have modified the abstract, introduction and discussion in line with the reviewer’s suggestions. Specifically, we no longer refer to our method as “simple” and we specifically highlighted that this method used “meiotic nuclei” in the Abstract. We also explicitly state in the discussion that our method cannot be used to study cytoplasmic mRNA (page13, lines 354-356). In addition, we now discuss the potential for our method to study transcriptomics, in light of recent advances in RNA-Seq from nuclei (page 13, lines 357-359).

Reviewer #2 (Remarks to the Author):

We would like to thank the reviewer for their comments and suggestions.

In this manuscript, the authors present a new method for retrieving highly specific population of male germ cells in the substages of meiosis I prophase. They show the utility of this with an extensive ChIP-seq analysis profiling histone modifications throughout meiotic prophase. The findings enlarge our understanding of the much-studied histone H3K4me3 marks at both genetic recombination sites and promoters, and also reveal considerable complexity throughout meiotic prophase of other, less well-studied histone marks. Most importantly, this work provides a method that can be exploited for many other studies of dynamic transitions during meiotic prophase.

Premise: For decades, researchers in the field of mammalian meiosis have attempted to refine and scale methods for the study of chromatin and cellular dynamics during meiotic prophase, especially in the male, where its study is experimentally more tractable than in the female. Prophase substages are defined cytologically and are a continuum rather than abrupt or static stages, a fact well reflected in gene expression data. There are no marker proteins unique to single substages, although several recent studies have successfully exploited combinatorial arrays of markers. Moreover, although spermatocytes increase in cell size, size differences are an imprecise means for sorting the substages. These features define the challenges to temporal mapping of meiotic prophase events that were tackled by the authors, and are set up well in the introduction.

Methods: Key to the nuclear FACS strategy was a set of reliable intra-nuclear protein antibodies, no one of which was unique to a single substage, but which in combinatorial arrays could discriminate the sub-stages of meiotic prophase. These were standard markers that have been used in other studies to define substages, but this is seemingly the first time that they have been in a strategy to sort nuclei. The results were impressive: reasonable purity and the cells look at least somewhat like what they are supposed to be by classical cytological criteria (Suppl. Fig. 1). The modifier "somewhat" is used because the cells are not optimally spread; most likely the prior fixation renders these preparations more similar in resolution and detail to whole mounts than to spread chromatin, a caveat that the authors might want to add.

R2.1: Our protocol does not allow for the preparation of spermatocyte spreads. Nonetheless, we can confidently stage nuclei after our fixation protocol, consistent with studies that use structurally preserved nuclei. Potentially, this new approach to imaging may allow the study of structures that are disrupted by spreading.

In view of the objectives of the present study, this is a minor concern and in any case, probably not resolvable at this time; nonetheless it does pose a cautionary note that these preparative procedures, with initial fixation (crosslinking), may not be suitable for all studies of meiotic chromosome dynamics. Encouragingly however, cells were obtained in sufficient number for standard genomic protocols.

R2.2: We have highlighted in several places that our nuclei sorting relies on initial fixation.

Results: The meiotic prophase sub-stage-specific cell populations were subjected to ChIP-seq to detect a number of histone modifications, with interesting, but not entirely conclusive, results and insights. For example, H3K4me3 marks at recombination hotspots were confirmed but temporal dynamics reveal that there is more to be learned about the removal of these marks as recombination proceeds.

For the most part, H3K4me3 marks at TSS correlated with RNA-seq data on transcription (even though the latter data were not as highly resolved temporally as the present analysis on histone marks). Perhaps most interesting with respect to H3K4me3 marks is that a substantial proportion of them occur at unannotated sites, clearly a world yet to discover.

Analysis of other histone marks at hotspots makes steps towards defining a histone code unique to hotspots. Of particular interest is the finding of a strong H3K9ac signal, demonstrating there may be more to this story than PRDM9. However, in spite of an exhaustive analysis, no definitive histone code was found to be clearly unique to or predict strength of hotspots. Indeed, it is known that there are more genomic PRDM9 sites activated than DSB sites, and more DSBs than there are reciprocal recombination events. Thus the epigenetic modifications responsible for this progressive selection are of major importance. This study points a way to resolve this issue.

Significance: In brief summary, both the method and the findings on histone modifications are of value to the community. The significance and impact of this paper will be considerable, and, in the authors' words, these studies "open the door" to avenues for better understanding of the molecular dynamics of chromatin in meiotic prophase I.

R2.3: Given the reviewer's interest in these aspects of our work, we would like to highlight that some changes were made to these sections of the manuscript. The rationale for these changes is outlined in the general response section.

Reviewer #3 (Remarks to the Author):

We would like to thank the reviewer for their comments and suggestions. We have performed a comprehensive series of experiments to address the reviewer's major concern regarding the quantification of ChIP-Seq signals.

General comments:

In this study, authors developed a new method to purify five specific population of meiotic cells from adult mouse testis by a relatively simple FACS sorting. Although this method seems to be limited to fixed nuclei thus cannot be utilized for RNA and native protein isolations, this long-awaited method has a potential impact on the study of meiosis especially using mammalian species. Authors demonstrated changes in cell-type specific histone modification during MPI, some of which are associated with hotspots and gene promoters. On the other hand, unfortunately, the biological findings demonstrated by the authors are still within the scope of previous findings and lack the novelty.

In addition, I also have several concerns regarding both technical and writing aspects described as follows;

Major comments:

1) *"Introduction" was poorly described; there was little information about the history of this research field and what previous studies have done. Instead, there were lots of description about the results and discussion of this study, so it's just like an another "abstract".*

R3.1: We have modified the introduction to accommodate the reviewer's suggestions.

2) *For data normalization between different samples, authors utilized a method called "NCIS" proposed by Liang et al. in 2008. This method is well-known, but not popular in the field. Although authors seemed to normalize the inter-sample variations carefully and manually, I'm not still certain that their normalization was appropriate, especially because they observed the most prominent H3K4me3 peaks at hotspots in zygotene, while PRDM9 reaches the highest level in leptotene according to the previous studies. In Fig. 3B, zygotene exhibited a noisy background, while others didn't have such backgrounds. Does this affect the peak intensities in zygotene? Whatever the reason, I highly recommend to perform "Spike-in" for more precise normalization, particularly because this kind of study will be recognized as a standard, and many researchers in the field will possibly refer in the future.*

R3.2: We agree with the reviewer that some assumptions underlying our normalization may have had room for error. Therefore, we have performed an extensive series of experiments using “spike-in” controls for H3K4me3. In summary, we found that the initial normalization was accurate and that H3K4me3 at hotspots is, indeed, maximal at zygotene. To expose these changes to all reviewers, the details are outlined in the general response section.

3) The authors made some discussion about the balance among DSBs, H3K4me3 and PRDM9 in the “Result” section (page 8, line 189-196) without showing data. Because this point is very important and substantial data are essential, I suggest to employ Prdm9 KO cells to verify their thought.

R3.3: This text in the results section was intended as discussion and not as a statement of results. This extensive discussion has been removed from the results section. We do not think that experiments in Prdm9 KO cells would illuminate aspects of DSB repair dynamics in meiosis. Thus, we have not performed experiments following this line of logic.

4) In the paragraph of page 8 line 202-224, the description sometimes unmatched to the Fig. 3D. First of all, authors described that “We found that the H3K4me3 profiles at TSSs were positively correlated with gene expression through MPI (Fig. 3D, $R = 0.32$; Spearman test)”, while I do not see such data in the figure.

R3.4a: We apologize for the incorrect figure citation. These data are not displayed on the figure. The R-value here refers to the overall correlation between the H3K4me3 profiles at TSSs and the gene expression before clustering. We have now corrected this in the text (page 8, line 190).

Second, authors mentioned that “This is particularly evident in cluster 1, where many genes have higher mRNA levels late in MPI than we would predict from H3K4me3 data (Supplementary Fig. 3). This may be explained by mRNA accumulation through MPI, or by H3K4me3 marking poised, but not yet active promoters”. How do authors think that way, as there are decreased levels of H3K4me3 and significantly higher levels of transcripts in the late MPI?

R3.4b: Our logic is as follows: In cases where H3K4me3 is high in early MPI, but expression is low, this could result from H3K4me3 at “poised” promoters, where no active transcription is occurring. Once a promoter leaves this poised state to become transcriptionally active, transcripts begin to appear. If these transcripts are long-lived, they would accumulate through MPI and give a maximal signal at diplonema. This is just one possible explanation of the

disconnect we observe. This logic has been clarified in the revised manuscript (page 13, lines 349-353).

Furthermore, it would be interesting to see how many of H3K4me3 marks on the gene promoters are regulated by PRDM9. If they are not made by PRDM9, is there any candidate methyltransferase, such as SET1? In addition, are these H3K4me3 marks on gene promoter specific to meiocytes? How about in spermatogonia?

R3.4c: Although PRDM9 certainly binds near some promoters, it is difficult to ascertain how much of the H3K4me3 at these promoters is attributable to PRDM9. This is primarily because both PRDM9 and other mechanisms may result in H3K4me3. About 900 (of 76,000) transcript start sites have H3K4me3 signals in wild-type mice, but not in mice lacking Prdm9.

Nonetheless, cellularity differences between wild-type and knockout mice complicate this simple comparison. We have therefore not included this discussion in the manuscript.

We do not have H3K4me3 ChIP-Seq data on the stages that precede leptotene which would be required to answer the reviewer's question regarding spermatogonia. Little is known about histone modifiers that operate in meiosis therefore to identify the methyltransferase responsible for the majority of promoter H3K4me3 marks is a non-trivial exercise and beyond the scope of this study.

5) In the paragraph of page 9 line 226-251, authors explained and "discussed" about Fig. S4. Although readers may be able to assume, there was no indication of cluster names in the figure, and it's difficult for me to understand their explanation based on the exhibiting dataset due to their rough and unkind description.

R3.5a: We apologize for the missing labels and have corrected this on the revised figures.

Also in the same paragraph, authors claimed that these PRDM9-dependent H3K4 methylation varied during MPI and that many H3K4 methylation retained until zygotene. However, this is not surprising because only 200-300 DSBs can be formed during MPI. It would be more interesting to see if these H3K4 methylation marks are generated in a spatiotemporal manner. Use of Spo11-KO would be helpful.

R3.5b: The reviewer misunderstood our intention here, as this set of sites was intended only as a positive control for our clustering strategy. Using Spo11 knockouts for interrogating the temporal patterning of H3K4me3 at Prdm9-defined hotspots is problematic. This is because in Spo11 knockout mice, meiotic DSBs cannot form and meiocytes arrest at a zygotene-like stage.

These disruptions to both cellularity and to the temporal dynamics of DSB-associated marks negate the utility of Spo11-KO mice to study the “spatio-temporal” patterning of H3K4me3 at hotspots. For these reasons, we did not perform experiments using Spo11-KO mice.

6) In Fig. 4, S5, and S6, authors performed ChIP-seq using 17 antibodies of modified histones. I highly appreciate their effort, while according to the Fig. 3a and Table S1, the cell number for the ChIP-seq (16,000) seems insufficient for global peak detection (note that Fig. 3a was calculated for H3K4me3, and other antibodies have much less ChIP efficiency thus they may require more number of cells).

R3.6: In examining the 17 histone marks, we did not perform peak calling. We shared the reviewer’s concern regarding ChIP-Seq sensitivity, and explicitly address it in Supplementary Fig. 9. All experiments except H3K9me2 and H3K27me1 revealed signals (enrichment / depletion) at the expected functional genomic regions (Supplementary Fig. 9). Nonetheless, transient or infrequent histone modifications may still be missed, and we now explicitly address this in the revised manuscript (page 10, line 265). We have also removed Fig. 3a; as mentioned by the reviewer, it was only relevant to ChIP-Seq using the Abcam ab8580 antibody and in light of the compromised specificity of this antibody (see general response), it added little value.

7) Although it’s not directly linked to this study, comparison of meiotic cell properties between juvenile and adult appears to be interesting, as the first and subsequent round of spermatogenesis can be different, while no substantial studies have demonstrated this question.

R3.7: We thank the reviewer for this interesting idea. However, while our system certainly offers a novel way to address this question, such work is a study unto itself, and far beyond the scope of this paper.

Minor comments:

8) As also pointed out above, lots of speculation and discussion were included in the result section. This is very confusing.

R3.8: We have restructured the results and discussion to address the reviewer’s concern.

9) Supplementary Fig. 2c was referred in the manuscript (page 8, line 190), but the figure is missing.

R3.9: This panel has been removed from the modified manuscript. All labelling has been thoroughly checked in the revised manuscript.

Reviewers' comments:

Reviewer #3 (Remarks to the Author):

Major comment:

In the revised manuscript, authors carefully answered questions from the reviewers by adding new data and rewriting the manuscript. Many of their answers satisfy the questions. In particular, for precise normalization, authors performed new ChIP-seq experiment using spike-in control, and they claimed that majority of H3K4me3 peaks can be observed during leptotene and zygotene and diminished by early pachytene as a major updated finding.

However, although I missed this point in the first submission, I recognized a potential technical problem. The authors performed ChIP posterior to immunostaining of chromatin-bound proteins (Stra8, SCP3, SCP1, and H1t). After nuclear sorting, chromatin was digested and subjected to ChIP by adding the primary antibody followed by the addition of Protein G dynabeads (P. 26, line 609-617 for H3K4me3 ChIP. No description was found for other antibodies). In this case, it's theoretically possible that Protein G can capture IgGs used for immunostaining, and it can cause the background of ChIP-seq. How did the authors manage this problem and normalize the data between different meiotic stages? This issue seems critical and has to be clarified.

Minor comments:

Fig2-4, Abbreviation for substages needs to be shown consistently: Leptotene is sometime L and LE.

Fig3a, There is no description for lower panel showing violin plots. Similarly, Fig3b need more detailed description. What does HS stand for? Hotspots as in Fig4a?

Fig4a,b, The legend should indicate stage of sorted cells used for ChIP-seq.

Fig4c, Like Fig3a, there is no description for lower panel showing violin plots.

Response to reviewers' comments on manuscript NCOMMS-18-34243-A:

Reviewer #3 (Remarks to the Author):

Major comment:

In the revised manuscript, authors carefully answered questions from the reviewers by adding new data and rewriting the manuscript. Many of their answers satisfy the questions. In particular, for precise normalization, authors performed new ChIP-seq experiment using spike-in control, and they claimed that majority of H3K4me3 peaks can be observed during leptotene and zygotene and diminished by early pachytene as a major updated finding.

1). However, although I missed this point in the first submission, I recognized a potential technical problem. The authors performed ChIP posterior to immunostaining of chromatin-bound proteins (Stra8, SCP3, SCP1, and H1t). After nuclear sorting, chromatin was digested and subjected to ChIP by adding the primary antibody followed by the addition of Protein G dynabeads (P. 26, line 609-617 for H3K4me3 ChIP. No description was found for other antibodies).

R1: The ChIP procedures are the same for the other antibodies. We have now clarified it in the revised version (P. 28, lines 623-628).

2). In this case, it's theoretically possible that Protein G can capture IgGs used for immunostaining, and it can cause the background of ChIP-seq. How did the authors manage this problem and normalize the data between different meiotic stages? This issue seems critical and has to be clarified.

R2: In anticipation of this concern we performed a ChIP-Seq control using leptotene chromatin but without adding any antibodies for ChIP. The background signal of this control is similar to that in other H3K4me3-ChIP-Seq experiments and there is no enrichment at DSB hotspots. Furthermore, our normalization strategies are very robust to variations in the ChIP-Seq signal:noise ratio, independent of the source.

Nonetheless, since it remains possible that IgGs used for immunofluorescence staining remain bound to chromatin and contribute to the ChIP signal, we have added text to the revised methods to alert the reader to this possibility (P. 25, lines 542-545).

Minor comments:

3). *Fig2-4, Abbreviation for substages needs to be shown consistently: Leptotene is sometime L and LE.*

R3: We have now standardized all abbreviations used in Fig. 2-4.

4). *Fig3a, There is no description for lower panel showing violin plots. Similarly, Fig3b need more detailed description. What does HS stand for? Hotspots as in Fig4a?*

R4: We have revised the figure legend accordingly.

5). *Fig4a,b, The legend should indicate stage of sorted cells used for ChIP-seq.*

R5: We have highlighted the nuclei stage in the figure legend.

6). *Fig4c, Like Fig3a, there is no description for lower panel showing violin plots.*

R6: We have revised the figure legend.